# Nutraceutical Difference between Two Popular Thai Namwa Cultivars Used for Sun Dried Banana Products

**DOI:** 10.3390/molecules27175675

**Published:** 2022-09-02

**Authors:** Donporn Wongwaiwech, Sudthida Kamchonemenukool, Chi-Tang Ho, Shiming Li, Tipawan Thongsook, Nutthaporn Majai, Duangporn Premjet, Kawee Sujipuli, Monthana Weerawatanakorn

**Affiliations:** 1Department of Agro-Industry, Faculty of Science and Agricultural Technology, Rajamangala University of Technology, Lanna Tak, 41/1 Moo 7 Paholayothin Road, Mai Ngam, Muang, Tak 63000, Thailand; 2Department of Agro-Industry, Faculty of Agriculture, Natural Resources and Environment, Naresuan University, 99 Moo 9, Tha Pho, Phitsanulok 65000, Thailand; 3Department of Food Science, Rutgers University, 65 Dudley Road, New Brunswick, NJ 08901, USA; 4College of Life Sciences, Huanggang Normal University, Huanggang 438000, China; 5Departmant of Agricultural Science, Faculty of Agriculture, Natural Resources and Environment, Naresuan University, Phitsanulok 65000, Thailand; 6Center of Excellence in Research for Agricultural Biotechnology, Naresuan University, Phitsanulok 65000, Thailand

**Keywords:** flavonoids, phenolic acid, dopamine, inulin, antioxidant activities, banana product

## Abstract

Musa (ABB group) “Kluai Namwa” bananas (*Musa* sp.) are widely grown throughout Thailand. Mali Ong is the most popular Kluai Namwa variety used as raw material for sun-dried banana production, especially in the Bangkratum District, Phitsanulok, Thailand. The sun-dried banana product made from Nanwa Mali Ong is well recognized as the best dried banana product of the country, with optimal taste compared to one made from other Kluai Namwa varieties. However, the production of Mali Ong has fluctuated substantially in recent years, leading to shortages. Consequently, farmers have turned to using other Kluai Namwa varieties including Nuanchan. This study investigated the nutraceutical contents of two popular Namwa varieties, Mali Ong and Nuanchan, at different ripening stages. Nutraceuticals in the dried banana products made from these two Kluai Namwa varieties and four commercial dried banana products were compared. Results indicated that the content of moisture, total sugar, and total soluble solids (TSS) (°Brix) increased, while total solids and texture values decreased during the ripening stage for both Kluai Namwa varieties. Rutin was the major flavonoid found in both Namwa Mali Ong and Nuanchan varieties ranging 136.00–204.89 mg/kg and 129.15–260.38 mg/kg, respectively. Rutin, naringenin, quercetin and catechin were abundant in both Namwa varieties. All flavonoids increased with ripening except for rutin, gallocatechin and gallocatechin gallate. There were no significant differences (*p* < 0.05) in flavonoid contents between both varieties. Tannic acid, ellagic acid, gallic acid, chlorogenic acid and ferulic acid were the main phenolic acids found in Mali Ong and Nuanchan varieties, ranging from 274.61–339.56 mg/kg and 293.13–372.66 mg/kg, respectively. Phenolic contents of both varieties decreased, increased and then decreased again during the development stage. Dopamine contents increased from 79.26 to 111.77 mg/kg and 60.38 to 125.07 mg/kg for Mali Ong and Nuanchan, respectively, but the amounts were not significantly different (*p* < 0.5) between the two Namwa varieties at each ripening stage. Inulin as fructooligosaccharide (FOS) increased with ripening steps. Production stages of sun-dried banana products showed no statistically significant differences (*p* < 0.05) between the two Namwa varieties. Therefore, when one variety is scarce, the other could be used as a replacement in terms of total flavonoids, phenolic acid, dopamine and FOS. In both Namwa varieties, sugar contents decreased after the drying process. Sugar contents of the dried products were 48.47 and 47.21 g/100 g. The drying process caused a reduction in total flavonoid contents and phenolic acid at 63–66% and 64–70%, respectively. No significant differences (*p* < 0.05) were found for total flavonoid and phenolic contents between the dried banana products made from the two Namwa varieties (178.21 vs. 182.53 mg/kg and 96.06 vs. 102.19 mg/kg, respectively). Products made from Nuanchan varieties (24.52 mg/kg) contained significantly higher dopamine than that from Mali Ong (38.52 mg/kg). The data also suggest that the banana maturity stage for production of the sun dried products was also optimum in terms of high nutraceutical level.

## 1. Introduction

Bananas, a popular global fruit crop, belong to the genus Musa in the family Musaceae, order Scitamineae. They grow in the tropical and subtropical areas of more than 130 countries including Southeast Asia [1,2,3]. The Asian Continent is the world’s largest banana producer, accounting for 54.4% of global production. The majority of edible cultivars are allopolyploid triploids with a genome constitution of AAA (dessert banana), AAB (plantains) and ABB (cooking bananas) [1,4]. Almost all cultivated bananas can be classified using 15 morphological characteristics into different genome groups [5]. In Thailand, banana (*Musa* sp.) production was 1.36 million tons in 2020 (World Data Atlas). Musa (ABB group) “Kluai Namwa” is widely grown in different areas throughout Thailand. Mali Ong, a variety of Kluai Namwa, is the most popular variety used as raw material for sun-dried banana production, especially in Bangkratum District, Phitsanulok Province located in the lower northern region of Thailand. Banana planting is the major occupation of the people in Phitsanulok, and the sun-dried banana products produced in Phitsanulok have become well known as the best solar dried banana products of the country. The solar-dried banana products in Phitsanulok were originally produced from the Kluai Namwa Mali Ong variety. There are some opinion pieces saying that the color and taste of solar dried banana products made from Mali Ong are optimal compared with other Kluai Namwa varieties. However, Mali Ong production has fluctuated substantially in recent years, and farmers now use other Kluai Namwa varieties, especially Nuanchan. The consumer concerns about health benefits associated with phytochemicals in food are increasing, but there is no information available regarding the phytochemical composition and nutraceutical contents of Mali Ong and Nuanchan, while data on the physicochemical properties and nutraceutical contents of solar dried banana products made from these two varieties are also scarce.

With high nutritional value but low-fat content, Kluai Namwa is a good source of nutrients including minerals, vitamins and carbohydrates [6]. Kluai Namwa is also packed with nutraceuticals including flavonoids, phenolic compounds, phenolic acid, dopamine, norepinephrine and prebiotics [2,3]. Consequently, the sun-dried banana products are rich in nutritional value. Nutrient compositions of raw Kluai Namwa fruit and dried banana products are shown in Table 1.

Bananas have been reported as a good source of phenolic compounds, in particular the flavonoids [8,9]. Flavonoids, a large family of polyphenolic plant compounds with six major subclasses, are widespread in the human diet. Consumption of flavonoids helps to protect against cardiovascular disease [10]. Flavonoids exhibit a wide range of biological activities including strong antioxidant capacity, anti-inflammatory, anti-neoplastic, and hepatoprotective activity as well as vasodilatory actions [10,11,12,13]. Flavonols such as quercetin and kaempferol have the most potent effect on cardiovascular protection [14,15]. Various phenolic acids and flavonoids have been identified in bananas including ferulic, gallic, p-hydroxybenzoic, vanillic and syringic acids, p-coumaric acids, ellagic acid, catechin, epicatechin, quercetin, tannins, rutin and gallocatechin [3,11]. Kanazawa and Sakakibara [16] found dopamine (L-3,4-dihydroxyphenylalanine) levels ranging from 2.5–10 mg/100 g in banana pulp. The antioxidant capacity of dopamine was higher than BHA, BHT, flavonoids, glutathione and catechin and similar to the strong antioxidants gallocatechin gallate and ascorbic acid [16]. Oral administration of dopamine is known to relieve Parkinson’s disease, a progressive disorder associated with dopamine deficiency in the brain [17]. Fructooligosaccharide (FOS) is a partial enzymatic hydrolysis fraction of inulin that contains a short chain of inulin oligomers with DP < 10 [7]. FOS exhibits as a prebiotic by promoting the growth of beneficial microorganisms such as Bifidobacterium and Lactobacillus in the colon that limit pathogens and reduce the risk of diseases such as colon cancer and inflammatory bowel disease, reduce pathogenic microorganisms and increase immunity [7,18,19,20]. Among the various natural sources of inulin, banana is one of the most promising fruits [21,22]. Study results of inulin yielded various content in banana, dry mass matter including 4.32–6.02 g/100 g [22], 0.29–1.63 g/100 g [19], 1.49–3.19 g/100 g [18] and 9.73–40.17 g/100 g [21].

Chemical composition including nutrients and phytochemicals and the bioactivity of banana fruit is impacted by various factors including maturity stage, climate, and agronomic, genomic and postharvest conditions [23,24,25]. During fruit ripening, several biochemical, physiological, and structural modifications occur, which determines fruit quality attributes, especially the nutritional value [23]. Banana fruit is generally divided by maturity or ripening into seven stages of development as 1. all green, 2. green with a trace of yellow, 3. more green than yellow, 4. more yellow than green, 5. yellow with a trace of green, 6. all yellow and 7. all yellow with brown speckles [26]. The stage of maturation is considered when selecting material for application, depending on the purpose of its potential usage. Generally, harvesting banana fruits (Kluai Namwa) for sun-dried banana products is performed at the all green stage (Figure 1a). After incubated storage for a couple of days depending on weather conditions, banana fruits at the optimal stages of 5–7 are ready for producing the sun-dried products (Figure 1b). Many nutraceuticals in banana fruits such as naringin, rutin, norepinephrine and dopamine were reported to reduce during the ripening stage [16]. By contrast, the inulin and fructan contents of banana increased during ripening, indicating aggregation of these carbohydrate polymers [18,19,21].

The objectives of this study included nutraceutical exploration of two popular Kluai Namwa varieties as Namwa Mali Ong and Nuanchan at different ripening stages. Nutraceutical contents of the dried banana products made from these two Kluai Namwa varieties and also four commercial dried banana products were investigated.

## 2. Results and Discussion

### 2.1. Physicochemical Properties of Namwa Mali Ong and Nuanchan Varieties during the Ripening Stage

After six days of storage, physicochemical properties of Namwa Mali Ong and Nuanchan varieties were determined. The fast change in color of banana peel during ripening is attributed to a climacteric fruit with the ripening process triggered by ethylene [23]. Seven ripening stages were described by Soltani et al. [26] as stage 1 green banana, stage 2 green with a trace of yellow, stage 3 more green than yellow, stage 4 more yellow than green, stage 5 yellow with a trace of green, stage 6 all yellow and stage 7 all yellow with brown speckles. Pictures showing bananas (Mali Ong) at different days of the defined ripening stages are presented in Figure 2. Moisture content, total solids, total soluble solids (TSS), sugar content and texture of Namwa Mali Ong and Nuanchan varieties at different ripening stages are shown in Figure 3 and Figure 4.

Moisture contents of Namwa Mali Ong and Nuanchan varieties ranged in 63.98–69.80% and 64.71–71.90%, respectively. No statistical difference of moisture contents were observed between the two varieties during ripening. Moisture contents of both varieties significantly increased during the ripening stages, whereas total solids decreased with increasing storage time. This phenomenon concurred with Mohapatra et al. [27]. Moisture migrated from banana peel to the pulp and increased sugar content due to starch hydrolysis [27], with respiratory breakdown of starches [28].

TSS (°Brix), texture, and sugar contents of Namwa Mali Ong and Nuanchan varieties during the ripening stage are shown in Figure 4 and Figure 5, respectively. The TSS of Namwa Mali Ong and Nuanchan varieties increased from 2.8 to 30.0 °B and 2.5 to 28.4 °B, respectively, during six days of storage, respectively. No statistically significant differences were shown between the two varieties. Mohapatra et al. [27] reported an increase in TSS of banana from 5 to 23.4 °B during nine days of storage. Increase of TSS in banana during banana ripening is dependent on cultivars [29,30]. Sugar contents based on dry mass of Namwa Mali Ong and Nuanchan varieties during the ripening stage are shown in Figure 5. The major sugar in the two Namwa banana varieties was fructose, followed by glucose, and both sugar contents increased during the ripening stage. Sucrose was not detected in ripening stage 1 but increased for 2–3 days before becoming absent (not detected). Total sugar contents increased from 0.78 to 61.54 g/100 g and 0.89 to 57.99 g/100 g for Namwa Mali Ong and Nuanchan, respectively. Increase in total sugar contents during the ripening process corresponded to an increase in TSS (Figure 4), and the sugar contents of the two varieties (Figure 5) were slightly different. Sugar content of Namwa Mali Ong was higher than Namwa Nuanchan, especially the stage for the dried banana product production (D_3_ and D_4_ at Stages 4–6) as 34.25 and 34.25% for Mali Ong and 12.67 and 31.49% for Nuanchan, and consistent with TSS contents of 29 °B and 27 °B, respectively. Unripe bananas are good sources of starch (70–80%), whereas the amount of sugar increased with reduction in the starch content during the ripening stage due to the respiratory breakdown of starches [28].

The texture (N) of banana fruit softened with ripening, and a rapid loss in firmness was observed after the second and third days of storage (Figure 4). Rapid changes in biochemical and other properties, including TSS, sugar accumulation and moisture content, were the main causes of alteration in textural properties. Degradation of nutrients and increased moisture content of the pulp caused a reduction in strength of the peel fiber, resulting in the flushness of pulp, which ultimately reduced the firmness quality of banana fruit [31].

### 2.2. Flavonoid Contents of Namwa Mali Ong and Nuanchan Varieties during the Ripening Stage

Flavonols, flavanones and flavanols as quercetin, kampherol, rutin, naringenin, apiginin (flavone), catechin, gallocatechin and gallocatechin gallate contents are listed in Table 2. Flavonoids and carotenoids are pigments that contribute to the yellow banana color in both the pulp and peel during ripening [23]. Results showed that apiginin was not detected in banana at all stages. Flavonoid contents increased with ripening, whereas gallocatechin gallate and gallocatechin decreased during the ripening process, concurring with Bennett et al. [32], who reported that levels of gallocatechin in banana cultivars Figo, Terra, Mysore and Pacovan significantly decreased during storage. Dong et al. and Someya et al. [11,33] reported gallocatechin in banana, while Youryon and Supapvanich and Fatemeh et al. [24,34] reported high content of flavonoids in banana fruit at the mature stage, which then declined as ripening advanced. Different banana varieties gave different results. Rutin was the major flavonoid found in both Namwa Mali Ong and Nuanchan varieties, with contents ranging 136.00–204.89 mg/kg and 129.15–260.38 mg/kg, respectively. Rutin, naringenin, quercetin and catechin were the most abundant in both Namwa varieties. Dong et al. [11] also found quercetin, catechin and gallocatechin in banana, with quercetin as the predominant flavonoid at all development stages in both types of banana. Flavonoids detected in bananas included quercetin, myricetin, kaempferol, cyaniding, rutin, myricetin and naringenin [35]. Quercetin, the most abundant flavonoid in the diet, exhibits anti-inflammatory action and antioxidant potential with an average intake of 26 mg to 1 g/day from fruits, vegetables, teas, wines, grains, and seeds [36,37,38].

All flavonoids increased in the two Namwa varieties with ripening, except for rutin, gallocatechin and gallocatechin gallate which decreased. Kanazawa and Sakakibara [16] also reported decrease of rutin during the ripening stage ranging 125–195 mg/kg. Total flavonoids of Namwa Mali Ong and Nuanchan varieties significantly increased from 459.44 to 584.67 mg/100 kg and 524.27 to 602.41 mg/100 kg, respectively. Each flavonoid and total flavonoid contents of both varieties for the dried banana product production (D_3_ and D_4_ at stages 4–6) were the highest at 526.18–560.66 mg/kg and 484.71–652.59 mg/kg, respectively. At each ripening stage, each flavonoid and total flavonoid contents were not statistically significantly different (*p* < 0.05) between the two Namwa varieties, especially at the stage for production of sun-dried banana products. When one variety is scarce, the other could be used as a replacement for dried banana production. Health benefits of flavonoids include anti-inflammatory, anti-neoplastic, anti-cancer and hepatoprotective activities as well as strong antioxidant capacity [3,9,11,12]. The mechanism of antioxidant activity of flavonoids involves excited oxygen species or direct scavenging of oxygen free radicals, and also inhibition of the oxidative enzymes that generate these reactive oxygen species [39]. Dong et al. [11] found that soluble flavonoids play an important role in the antioxidant activity of banana. Many studies detected quercetin, ellagic acid, catechin and gallocatechin at all banana developmental stages [11,33]. Bennett et al. [32] reported that levels of total phenolics decreased after banana harvesting, especially in ‘Mysore’ (from 18.9 to 9.8 mg of GAE equivalent/g of dry weight pulp). However, catechin significantly increased in ‘Mysore’ (from 51.4 to 143.2 μg/100 g of dry weight pulp) and significantly increased in epicatechin in ‘Nanicao’ (from 193.5 to 459.8 μg/100 g of dry weight pulp), while naringin was not detected in any of the cultivars. Apart from their powerful antioxidant activities, rutin, naringenin, quercetin and catechin also displayed anti-viral, anti-bacterial, anti-carcinogenic, anti-inflammatory and anti-obesity properties [9,40,41].

### 2.3. Phenolic Acid Contents of Namwa Mali Ong and Nuanchan Varieties during the Ripening Stage

Phenolic acid contents including gallic, ellagic, ferulic, chlorogenic and tannic acids of the two Namwa varieties are presented in Table 3. The main phenolic acid was tannic followed by ellagic, gallic, chlorogenic and ferulic acids. Total phenolic acids of Namwa Mali Ong and Nuanchan varieties during the ripening ranged 274.61–339.56 mg/kg and 293.13–372.66 mg/kg, respectively. The trend of phenolic contents in both Namwa varieties during the development stage was the same, showing decrease, increase and then decrease. These results concurred with previous studies [11,35,42,43] that also identified tannic acid, ellagic acid, gallic acid, chlorogenic acid and ferulic acid in bananas. Ellagic acid level increased during maturation [11]. By contrast, tannic acid level decreased during the development of the ripening step [35]. This was explained by increased polymerization and inactivity of tannins [35,44,45].

Each phenolic acid and total phenolic acid contents of Namwa Mali Ong and Namwa Nuanchan at the stage of dried banana production (D_3_ and D_4_) were also high at 275.78–333.61 mg/kg and 313.70–372.66 mg/kg, respectively. Comparing ripening stages, each total phenolic acid content was not statistically significantly different (*p* < 0.05) between the two Namwa varieties, except at D_6_. Phenolic acid contents, like flavonoid contents, were similar and the two Namwa varieties could both be used for sun-dried banana production.

### 2.4. Dopamine and FOS Contents of Namwa Mali Ong and Nuanchan Varieties during the Ripening Stage

Dopamine, one of the catecholamines, is a strong water-soluble antioxidant identified in banana [16]. Dopamine contents of Namwa Mali Ong and Nuanchan varieties during the ripening stage are shown in Figure 6. Dopamine contents increased from 79.26 to 111.77 mg/kg and 60.38–125.07 mg/kg, respectively with no significant differences (*p* < 0.05) between the two Namwa varieties at each ripening stage. The highest dopamine content was found at D_6_ for Namwa Mali Ong (111.77 mg/kg) and Nuanchan (125.07 mg/kg) varieties. Dopamine contents of Namwa Mali Ong and Nuanchan varieties at D_3_ and D_4_ were 73.94–84.71 mg/kg and 69.84–108.66 mg/kg, respectively, with no statistically significant difference (*p* < 0.05) between two varieties. Data indicated that dopamine content increased with ripening development, consistent with Kanazawa and Sakakibara (2000) [16] who reported dopamine in banana pulp for ripened banana ready to eat at 25–100 mg/kg. Out of eight ripening stages, they also reported that dopamine in banana peel decreased with ripening, while dopamine in the pulp increased during ripening (from stage 1–stage 6) and then decreased [16].

Dopamine plays a key role as a neurotransmitter and precursor for norepinephrine and epinephrine and was found at high levels in banana peel and pulp [16]. Dopamine has higher antioxidant potency than glutathione, food additives such as butylated hydroxyanisole and hydroxytoluene, flavone luteolin, flavonol quercetin and catechin [16]. Dopamine deficiency can lead to depression, loss or decrease of motor control and lack of motivation to carry out normal routines. Increasing consumption of dietary dopamine directly helps to relieve these symptoms. Banana is a very good source of dopamine and L-tyrosine and can be used as a natural supplement to restore low dopamine levels [16,46].

The highest inulin contents in form of FOS of Namwa Mali Ong and Nuanchan varieties were 0.12 g/100 g and 0.75 g/100 g, respectively, at D_6_ (Figure 2). Results showed that inulin increased with ripening development as dopamine (Figure 7) from 0.05 to 0.12 g/100 g for Namwa Mali Ong and from 0.00–0.75 g/100 g for Namwa Nuanchan varieties. Compared with Namwa Mali Ong, Nuanchan varieties had a high FOS level. Tanjor et al. [7] reported FOS contents of unripe and ripe banana fruits at 0.06 and 0.25 g/100 g, respectively, indicating that FOS increased with banana ripening. Similarly, Pongmalai and Devahastin and Shalini and Antony [18,21] noted that fructan and inulin contents in banana increased during the ripening stage, indicating accumulation of these carbohydrate polymers. As a prebiotic, FOS exhibits various health benefits including stimulating the immune system, relieving constipation (15–20 g a day), decreasing the risk of osteoporosis, lowering the synthesis of triglycerides and cholesterol (8–10 g a day) and reducing plasma cholesterol concentrations [7].

### 2.5. Antioxidant Activities of Namwa Mali Ong and Nuanchan Varieties during the Ripening Stage

Antioxidant values including total phenolic contents (TPC), tannin, total flavonoid contents (TFC), DPPH scavenging ability (DPPH) and ferric reducing antioxidant power (FRAP) of both Namwa varieties are shown in Table 4. Antioxidant values increased during the ripening stage for both banana varieties. At D_5_, Namwa Mali Ong contained significantly higher antioxidant values than Nuanchan, except for DPPH value. In Mali Ong varieties, TPC increased from 160.01 to 601.38 mg GAE/100 g during ripening, consistent with DPPH (119.44 to 847.37 mg TE/100 g), and FRAP (970.28 to 5280 mg FeSO_4_/100 g) (Table 4 and Appendix A). In Nuanchan varieties, TPC increased from D_0_ (207.30 mg GAE/100 g) to the maximum level (728.57 mg GAE/100 g) at D_2_, then decreased and increased again after D_3_ in accordance with TPC, TFC, DPPH and FRAP (Appendix A).

Bashmil et al. [35] identified 24 polyphenols using the analytical technology of LC-ESI-QTOF-MS/MS and determined high antioxidant capacity of various varieties of banana pulp for scavenging free radicals by FRAP, DPPH, ABTS, FICA, RPA, OH-RSA and TAC methods. Generally, phenolic content increased as the banana ripened and then decreased slightly when the fruit became over ripe [23]. Other researchers found similar variations in TPC between different banana cultivars. Balasundram et al. [47] reported that phenolic content in Nanicao slightly increased during ripening, while TPC in other cultivars such as Figo, Terra, Mysore and Pacovan decreased during storage. Fatemeh et al. [24] also found that phenolic contents in ripe ‘Dream’ banana pulp flours were higher than in green Dream banana pulp flours, while Youryon and Supapvanich [34] reported that TPC content and FRAP of mature green Leb Mue Nang were significantly lower than both ripe and overripe fruit, but TPC was not significantly different. By contrast, amounts of TFC and DPPH radical scavenging activity were high at the mature stage and then declined as ripening advanced [11,24,35]. These results showed that changes in TPC, TFC, DPPH and FRAP varied depending on the cultivar, method of extraction and analysis technique.

### 2.6. Nutraceutical Contents of Commercial Dried Banana Products and Dried Banana Products from Namwa Mali Ong and Nuanchan Varieties

#### 2.6.1. Flavonoid Contents of Commercial Dried Banana Products and Dried Banana Products from Namwa Mali Ong and Nuanchan Varieties

Banana fruit at D_3_, stages 4–6 and D_4_, stage 7 of the two varieties (Figure 1 and Figure 2) were selected for dried banana production. The central part of the pulp, which is the consumable portion of the banana fruit, is highly nutritious [35]. Sun dried banana products are made from the whole banana fruit after the peel has been removed; therefore, sun-dried banana products are a rich source of nutrients and phytochemicals. The four commercial sun-dried banana products and the dried banana products made from Namwa Mali Ong and Nuanchan varieties, as shown in Figure 8a,b, were determined for sugar, nutraceutical contents and antioxidant activities. Total sugar contents including glucose and fructose decreased after the drying process in both Namwa varieties, as shown in Table 5. The main sugars in dried banana products were fructose and glucose. Total sugar contents of products from Namwa Mali Ong and Nuanchan varieties decreased 23% and 35%, respectively. There was no significant difference in sugar contents of dried banana products made from the two Namwa varieties (48.47 vs. 47.21 g/100 g, respectively). Sugar contents of the commercial dried banana products ranged 47.98–40.00 g/100 g.

Flavonoid contents including quercetin, kampherol, rutin, naringenin, catechin and gallocatechin of the dried banana products made from the two Namwa varieties were compared with the commercial products, as shown in Table 6. The drying process for solar dried banana caused total flavonoid contents of the products from Namwa Mali Ong and Nuanchan varieties to significantly decrease by 66% and 63%, respectively (Table 6). The loss of each type of flavonoid after the drying process varied from 21–77% for products made from Mali Ong and from 47–75% for Nuanchan varieties. There was no significant difference (*p* < 0.05) in total flavonoid contents between dried banana products made from the two Namwa varieties (178.21 vs. 182.53 mg/kg). Total flavonoid contents of the dried banana products ranged from 148.44–197.16 mg/kg, respectively with rutin, naringenin and catechin as the major constituents. The major flavonoids of the dried products were consistent with the study on flavonoid contents of Namwa Mali Ong and Nuanchan varieties during the ripening stage.

#### 2.6.2. Phenolic Acid Contents of Commercial Dried Banana Products and Dried Banana Products from Namwa Mali Ong and Nuanchan Varieties

Phenolic acid contents including tannic acid, gallic acid, ellagic acid, ferulic acid and chlorogenic acid of the dried banana products made from the two Namwa varieties compared with the commercial products are shown in Table 7. The total phenolic contents of dried banana products made from Namwa Mali Ong and Nuanchan varieties reduced by 70% and 64%, respectively, due to the drying process. There was no significant difference (*p* < 0.05) in total phenolic acid of dried banana products made from the two Namwa varieties (96.06 vs. 102.19 mg/kg). Phenolic acid contents of commercial dried banana products ranged from 88.49–126.06 mg/kg. Ellagic and tannic acid were abundant in dried banana products (26.94–46.29 mg/kg and 21.82–46.49 mg/kg, respectively) while ferulic acid was found in minor amounts ranging from 0 to 22.20 mg/kg. The loss percentage of each phenolic by the drying process was more than 50% (Table 7). Decreases in total phenolic compounds during the drying process were in agreement with previous studies. The polyphenolics were unstable and prolonged heat treatment caused irreversible chemical changes in the phenolic compounds [48,49,50]. Julkunen-Tiitto and Sorsa [51] reported loss of purple willow flavonoids and tannins as a result of drying treatments. de Ancos et al. [52] suggested that decrease of anthocyanins and other polyphenolics during drying was attributed to many factors other than heat treatment. These included the activity of organic acids, polyphenol oxidase, pH, oxygen and sugar concentration. During long thermal treatments in an open dryer, higher susceptibility to oxidation was observed due to the presence of heat and oxygen favoring enzymatic activity of polyphenol oxidase [50,53].

#### 2.6.3. Dopamine and Inulin Contents of Commercial Dried Banana Products and Dried Banana Products from Namwa Mali Ong and Nuanchan Varieties

Dopamine contents of the products made from Mali Ong and Nuanchan varieties were significantly reduced by 75% and 45%, respectively, as a result of the drying process (Table 8). Dopamine contents of the dried banana products ranged from 22.26 to 38.52 mg/kg and products made from Nuanchan varieties (24.52 mg/kg) contained significantly higher dopamine than Mali Ong (38.52 mg/kg). Fruits of the Musa genus, such as bananas, plantains and avocado, were also reported to contain high levels of dopamine at 4–5 mg/kg [54]. Data indicated that the dopamine content of solar dried banana products was higher than that of avocado. Inulin contents of the four commercial dried banana products ranged from 0.11 to 0.24 g/100 g (Figure 9). The average inulin content (dry mass) of the dried banana products was 0.18 g/100 g. When compared with inulin content of banana fruit during the ripening stage, the result implied that the drying process decreased inulin content by 25%.

#### 2.6.4. Antioxidant Activities of Dried Banana Products from Commercial Source and Namwa Mali Ong and Nuanchan Varieties

TPC, tannin and TFC values of banana significantly reduced after the drying process in both Namwa varieties. Losses of TPC, tannin and TFC by the drying process were 81, 79 and 76%, respectively, for products made from Mali Ong and 66, 61 and 59%, respectively for products made from Nuanchan varieties (Table 9). There were no significant differences in TPC, DPPH and FRAP values of banana products made from Namwa Mali Ong and Nuanchan varieties. TFC values of products made from Namwa Mali Ong were significantly higher than those made from Namwa Nuanchan varieties (120.79 vs. 105.54 mg in QE/100 g), but total flavonoids including quercetin, kampherol, rutin, naringenin, catechin and gallocatechin in Table 6 were not significantly different (*p* < 0.05) (178.21 vs. 182.53 mg/kg). Significant differences of TFC values between products from the two different Namwa varieties and the four commercial products resulted from the storage periods of the commercial one compared with freshly prepared products.

TPC, tannin, TFC, DPPH and FRAP values of the dried banana products ranged 193.47–286.62 mg GAE/100 g, 4.04–5.88 mg TAE/100 g, 44.71–59.16 mg QE/100 g, 617.37–643.85 mg TE/100 g and 1,110.58–1,151.72 mg FeSO_4_/100 g, respectively. Pearson’s correlation coefficient analysis (Appendix A) revealed strong positive correlation among TPC, tannin, FRAP, total phenolic acid, and total flavonoid contents, indicating that TPC, tannin, total phenolic acid and total flavonoid contents contributed more to antioxidant activity as FRAP. However, DPPH negatively correlated (−1) with the other values, while they strongly positively correlated with each other. No significant differences between the DPPH values of all solar dried products gave negative correlation of DPPH with the other values. Our results indicated that bananas had antioxidant activities equivalent to melons, strawberries, cauliflowers, apples (Fuji, Red Delicious, Gala and Liberty), blueberries (Rabbiteye, Southern highbush), guavas (white), red grapes, starfruit (sweet) and dried fruits such as apple, apricot and peach [47,55,56,57,58]. Antioxidant values of banana products made from the two different Namwa varieties were higher than the four commercial products that had been stored for a long time.

## 3. Materials and Methods

### 3.1. Materials

All chemicals used were analytical grade (for HPLC) and provided by Poch (Polish Chemical Reagents, Gliwice, Poland) or Merck (Darmstadt, Germany). Inulinase (I2017 inulinase from *Aspergillus niger*, CAS Number 9025-67-6 with enzyme activity of 1740 INU/G), Trolox (6-hydroxy-2,5,7,8-tetramethylchroman-2-carboxylic acid), standard gallic acid, DPPH (2,2-diphenyl-1-picrylhydrazyl, TPTZ (2,4,6-Tris(2-pyridyl)-s-triazine) and ABTS (2,2′-azinobis-(3-ethylbenzothiazoline-6-sulfonic acid) was purchased from Sigma-Aldrich (St. Louis, MO, USA). Sodium acetate trihydrate was purchased from Laboratory Reagents and Fine Chemicals (Mumbai, India). Glacial acetic acid was purchased from RCI Labscan (Bangkok, Thailand). Ferric chloride (lron lll) chloride 6-hydrate pure was purchased from AppliChem PanReac ITW Companies (Barcelona, Spain).

#### 3.1.1. Experimental Design of Banana Ripening Stage

Two varieties of green Kluai Namwa banana (ABB) (*Musa sapientum* L.) of Mali Ong and Nuanchan varieties were provided by a farm of Maesom Enterprise (small and medium-sized enterprise) (Bang Krathum, Phitsanulok, Thailand) and immediately transferred to our laboratory (Phitsanulok). The harvesting period were 110 and 120 days after flowering for Mali Ong and Nuanchan varieties, respectively. The harvesting process was performed by Associated Professor Dr. Duangporn Premjet (botanist) and a farmer. Banana plantlet from Maesom farm were obtained from Pant Propagation Center No. 6 (Phitsanulok). Clusters of each banana variety were separated into individual bunches of bananas and stored at room temperature (30 ± 0.81 °C) with relative humidity 40 ± 5.06%.

#### 3.1.2. Experimental Design of Solar Dried Banana Products

The two varieties of Kluai Namwa bananas at storage days 3–4 (D_3_ and D_4_ in Figure 2) and maturity 4–7 (Figure 1b) were selected for processing of dried banana products. Briefly, banana fruit at storage days 3–4 was peeled and dried in a parabolic dome at 50–60 °C for 4–5 days (Figure 10). The dried banana was then trimmed to remove defects, dried in a hot air oven at 70 °C for 30–45 min, vacuum packed, and stored at −20 °C for further investigation. Four well-known commercial sources of dried banana products in Phitsanulok Province, Thailand, including DBPA, DBPS, DBPM and DBPJ, were selected and their nutraceutical contents were compared (Figure 8a).

### 3.2. Moisture, Texture, Total Soluble Solids and Sugar

Moisture content of the samples was measured according to the AOAC method (2005; method 930.15). Total soluble solids were measured using a hand refractometer (HR-130, Optika, Italy) as °Brix after the samples were crushed by a grinder (BL43P, Tefal, China). Firmness of a whole banana sample without peel was determined by a texture analyzer (TA.HD plus, Stable Micro System, UK) using a needle probe number P/2N. The result was reported as peak force in Newtons (N) [59,60]. Total sugars as fructose, glucose and sucrose were determined by HPLC following the methods of Xu, Niimi and Han [61].

### 3.3. Flavonoid Content and Phenolic Acid Analysis by HPLC/DAD/MS

#### 3.3.1. Extraction Method

Identification and quantification of banana flavonoids and phenolic acid were adapted from the method by Tian, Nakamura, and Kayahara [62]. One gram of sample was dissolved in 10 mL methanol (100%) and vigorously shaken for 10 min. The mixture was filtered and centrifuged for 10 min at 4000 rpm. The extractions were performed four times. The methanol content of the combined methanolic extract was removed to obtain a 10 mL sample for HPLC analysis. Standard solutions of the phenolic compounds were prepared in methanol [62]. For gallocatechin gallate, gallocatechin, and naringenin, samples were dissolved in 20 mL of methanol (50%), incubated at 60 °C for 1 h, and sonicated for 30 min at room temperature. All sample solutions were filtered through 0.45 nylon filter membrane before injecting them to HPLC.

#### 3.3.2. HPLC-DAD and MS Detection Conditions

The analysis of flavonoids and phenolic acid were performed on an Agilent 1100 series HPLC system (Agilent Technologies, Böblingen, Germany) equipped with a diode array detector and an MSD SL quadrupole mass spectrometer with an API-ES interface. The phenolic compounds in the sample were estimated by a reversed-phase (a 150 mm 4.6 mm i.d., 5 um, LiChroCART RP-18e) column (Purospher STAR Merck, Darmstadt, Germany). The column was kept at 40 °C. The mobile phase consisted of a binary solvent system using water added with 10 mM of ammonium formate buffer pH 4 (solvent A) and 100% acetonitrile (solvent B), and a flow rate was kept at 1.0 mL/min. The gradient elution program was 0–5 min, 0% B; 5–10 min, 0–20% B; 10–20 min, 20% B; and 20–60 min, 20–40% B. All samples were filtered through a 0.45 nylon filter membrane. The injection volume was 20 μL. The chromatograms were obtained at a wavelength of 270 nm for analysis of phenolic compounds. The phenolic compounds were analyzed in positive ion modes with the following settings: capillary voltage 3500 V (Negative) and 4000 V. (positive). The nebulizer gas (N2) pressure was 60 psi; dry gas flow, 13 L/min; dry temperature, 320 °C. Analysis was carried out using scans from *m*/*z* 100 to 700 profile mode with a step size of 0.2. Flavonoids and phenolics were identified based on their retention times of commercial standards. Quantification of each compound was accomplished by comparing the peak areas with that of a calibration curve of each standard [63].

#### 3.3.3. HPLC-DAD and MS Detection Conditions for Gallocatechin Gallate, Gallocatechin, and Nariginin

The analysis was performed as the same condition in Section 3.3.2 with different conditions of mobile phase. The mobile phase consisted of a binary solvent system using 100% methanol (solvent A), 10 mM ammonium formate buffer pH 4 (solvent B) and kept at a flow rate of 1.0 mL/min. The gradient elution program was 0–5 min, 100% B; 5–10 min, 100 to 80% B; 10–20 min, 80% B; 20–50 min, 80 to 65% B.

### 3.4. Dopamine Analysis

#### 3.4.1. Extraction of Dopamine

The dopamine analysis was adapted from Seremet et al. [64]. Sample (1.5 g) was dissolved in 20 mL DI water and vigorously shaken for 10 min following incubation at 80 °C for 1 h. After cooling to room temperature, methanol of 20 mL was added and then sonicated for 60 min at 45 °C, and left at room temperature overnight (12–16 h). The mixture was then filtered and centrifuged for 10 min at 3500 rpm. The extracts were stored at −4°C until further analysis.

#### 3.4.2. HPLC and Detector Condition

The HPLC analysis was performed on an Agilent Series 1100 chromatographic system (Agilent Technologies, Santa Clara, CA, USA) using a Zorbax Extend C18 (4.6 × 250 mm, 5 µm i.d.) chromatographic column (Agilent Technologies, USA) and coupled with a Diode Array Detector (Agilent Technologies, Santa Clara, CA, USA). The elution was performed in a gradient with a two-component mobile phase consisting of (A) 0.1% (*v*/*v*) formic acid solution in methanol and (B) solution of 10 mM ammonium formate adjusted pH to 4 with formic acid, according to the following regimen: 0 min—20% A, 80% B; 20 to 30 min—50% A, 50% B. The flow rate was 1.0 mL/min, the injection volume 20 µL and the column temperature 25 °C. The chromatograms were recorded at 280 nm. Dopamine identification was performed by comparing the retention time and characteristic absorption with a commercially available standard. Quantification was accomplished by comparing peak area of a dopamine calibration curve (25–300 µg /mL) [64].

### 3.5. Inulin-Type Fructans and Fructoolisaccharide Determination

Inulin-type fructans, FOS and sugar were extracted following method 997.08 of the AOAC and determined by gas chromatography [7,65]. They were first extracted with hot water and then hydrolysed by inulinase and derivatized by oxymation and silylation reaction. Individual sugars were then determined by high-temperature gas chromatography [7]. For inulin extraction, freeze-dried samples containing 1 g of inulintype fructans were extracted with hot water at 60 °C. The pH of the solution was adjusted to 6.5–8.0 and brought to a final volume of 50 mL with deionized water, followed by incubation at 85 °C for 15 min. The extracted sample solution was mixed with acetate buffer (pH 4.5), and the pH was adjusted with acetic acid and Na-acetate to 4.5. A sufficient amount of inulinase was added and incubated for 30 min at 60 °C. The mixture was derivatized for GC analysis. FOS in hot water extract was determined and quantified from each calibration curve of GF2, GF3 and GF4. Sugar and FOS (GF2, GF3, GF4) were determined by high temperature GC (Agilent1 7890, Santa Clara, CA, USA), equipped with a capillary aluminium-clad column (length 6 m, ID 0.53 mm, coated with 5%-phenyl-polycarboranesiloxane, HT-5, Restek, Bellefonte, PA, USA). Temperature of the FID detector was set at 447 °C. Helium was used as the carrier gas at a flow rate of 9 mL/min. Air and hydrogen gas for the FID detector were set at 400 mL/min and 40 mL/min, respectively.

### 3.6. Total Phenolic Contents (TPC) and Antioxidant Activities Analysis

#### 3.6.1. Sample Extraction

Sample extraction for total phenolic contents (TPC) and antioxidant activities analysis was modified following the method of Youryon and Supapvanich [34]. Samples (2.5 g) were placed in 50 mL centrifuge tubes, and 15 mL methanol (80%) were added. The mixtures were homogenized by a homogenizer (T25, IKA, Staufen, Germany) at 6000 rpm for 2 min, and then the volumes were adjusted to 25 mL with methanol before filtering through filter paper No. 4. The filtered solutions were stored at −20 °C for further analysis.

#### 3.6.2. Total Phenolic Contents (TPC)

TPC was determined by the Folin–Ciocalteu Reagent (FCR) colorimetric method following Luque-Rodríguez et al. [66]. A 0.5 mL aliquot of sample extract was mixed with 1 mL of FCR using a vortex mixture. After 5 min, 2 mL of sodium carbonate solution (7.5%) were added, and the reaction mixture was incubated for 2 h at 37 °C. The absorbance was then measured by a spectrophotometer (Genesys 20, Thermo, Waltham, MA, USA) at 765 nm using gallic acid as the standard. Total phenolic contents were expressed as milligrams of gallic acid equivalent per 100 g of sample.

#### 3.6.3. Total Flavonoid Contents (TFC)

TFC were estimated spectrophotometrically as described by Yang et al. [67]. Briefly, 1.5 mL of extract and 0.075 mL of 0.5% sodium nitrite were mixed and allowed to settle for 6 min. Then, 0.15 mL of 10% AlCl_3_ were added and settled for another 5 min. Finally, 0.5 mL of 4% NaOH (1 M) and 0.775 mL of water were added and left for 15 min. The absorbance was measured at 510 nm using a UV-vis spectrophotometer (Genesys 20, Thermo, Waltham, MA, USA). TFC was expressed in mg of rutin equivalent (RE)/100 g DM.

#### 3.6.4. Tannin Content

Tannin content was determined following the methods of Espinosa and Santacruz and Aboul-Enein et al. [68,69] using tannic acid as a standard. Briefly, a fresh sample (25 g) was added to 80% methanol (25 mL) and then vortexed, followed by centrifugation (5 min, 6000 rpm). The supernatant was filtrated by a syringe filter. A 500 µL aliquot of the extract and 1 mL of Folin’s reagent were then added to a glass vial and incubated in the dark for 5 min (room temperature). Sodium carbonate solution (7.5% (*m*/*v*)) was added to the mixture and incubated for another 90 min. The absorbance was then read at 775 nm, with results expressed as mg of tannic acid equivalent per 100 g (mg TAE/100 g) DM.

#### 3.6.5. DPPH Free Radical Scavenging

2,2-Diphenyl-1-picrylhydrazyl radical scavenging ability (DPPH) of banana extracts were analyzed using spectrophotometric methods as described by Maier et al. [70]. DPPH working solution was freshly prepared at a concentration of 0.1 mM in absolute methanol. Then, 0.5 mL of extract and 1.5 mL of DPPH solution were mixed using a vortex (ZX4, Velp Scientifica, Usmate Velate, Italy) and incubated for 30 min. Absorbance of the mixture was measured at 517 nm using a UV-vis spectrophotometer (Genesys 20, Thermo, Waltham, MA, USA). Trolox was used as a standard, and DPPH radical scavenging activity was expressed as mg Trolox equivalent (TE)/100 g DM.

#### 3.6.6. Ferric Reducing Antioxidant Power (FRAP)

FRAP was determined using the colorimetric method, as described by Benzie and Strain (1996) [71]. Reagent A: 300 mM acetate buffer (pH 3.6), Reagent B: 10 mM TPTZ in 40 mM HCl and Reagent C: 20 mM FeCl_3_.6H_2_O were prepared and mixed (ratio 10:1:1) as FRAP reagent. Then, 0.2 mL of the extract were mixed with 2.0 mL of FRAP solution and left for 30 min in the dark. Absorbance of the mixture was read at 593 nm using a UV-vis spectrophotometer (Genesys 20, Thermo, USA). Ferrous sulfate was used as the standard, and the FRAP was expressed as mg FeSO_4_ E/100 g DM.

### 3.7. Statistical Analysis

All experiments were conducted in triplicate. Differences between means were determined by one-way analysis of variance (ANOVA) and Duncan’s post-hoc test (using SPSS statistical software version 23, IBM Corp., New York, NY, USA) with significance level *p* < 0.05.

## 4. Conclusions

Rutin was the main flavonoid followed by naringenin, quercetin and catechin in both Namwa banana varieties, and their contents were not significantly different at each ripening stage. Tannic acid followed by ellagic acid, gallic acid, chlorogenic acid and ferulic acid were the major phenolic acids found in both Namwa varieties. All flavonoids increased with ripening except for rutin, gallocatechin and gallocatechin gallate, while phenolic acid contents of both varieties decreased, increased and then decreased again during the development stage. The drying process for solar dried banana production caused a loss of nutraceuticals in banana including total flavonoid contents, phenolic acid and dopamine. No difference was recorded between total flavonoid and phenolic contents of dried banana products made from the two Namwa varieties. Dopamine content of solar dried banana products was high. Comparing the production stage of sun-dried banana products, the two Namwa varieties could be used as replacements in terms of total flavonoids, phenolic acid, dopamine and FOS when one variety is in short supply.

## Figures and Tables

**Figure 1 molecules-27-05675-f001:**
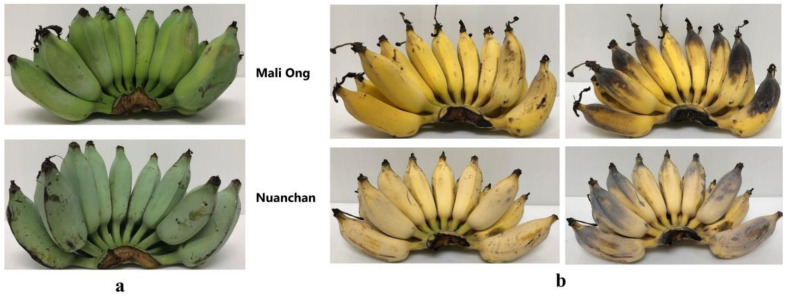
Kluai Namwa Mali Ong and Nuanchan varieties at (**a**) harvesting stage and (**b**) optimal stage for production of sun-dried banana products.

**Figure 2 molecules-27-05675-f002:**
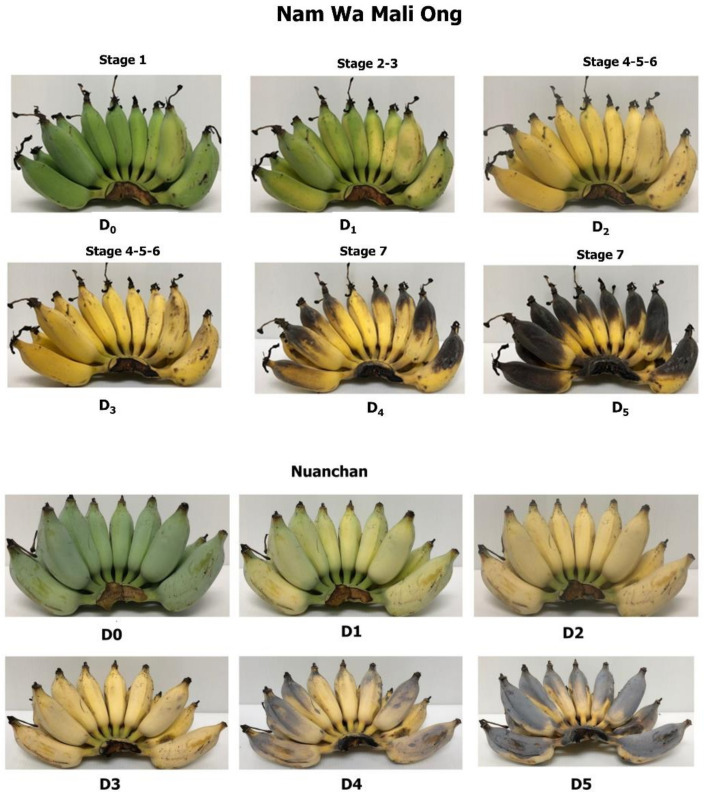
Stored banana at different days showing the seven ripening stages described by Soltani et al. [26].

**Figure 3 molecules-27-05675-f003:**
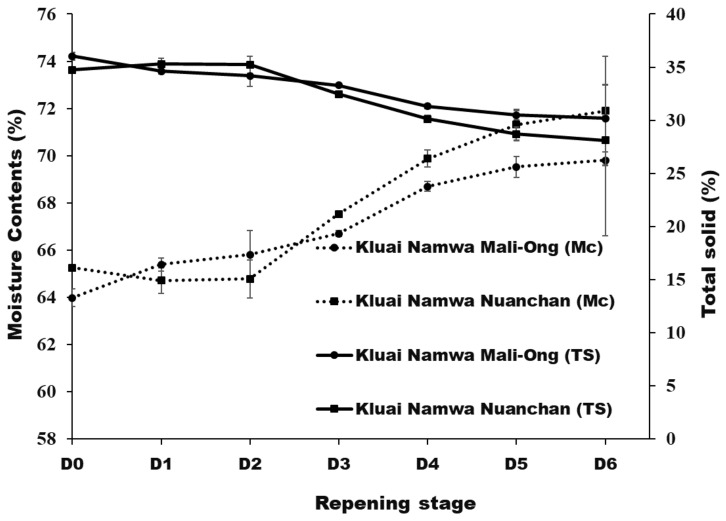
Moisture contents and total solids of Namwa Mali Ong and Nuanchan varieties during the ripening stage.

**Figure 4 molecules-27-05675-f004:**
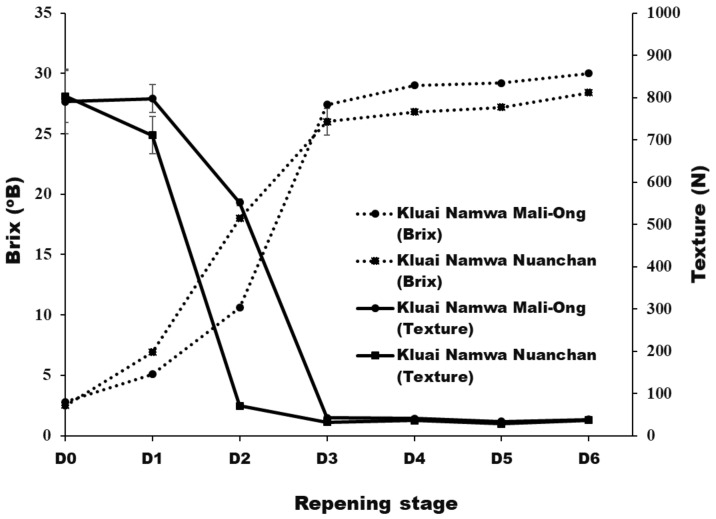
Total soluble solids (TSS) (°Brix) and texture of Namwa Mali Ong and Nuanchan varieties during the ripening stage.

**Figure 5 molecules-27-05675-f005:**
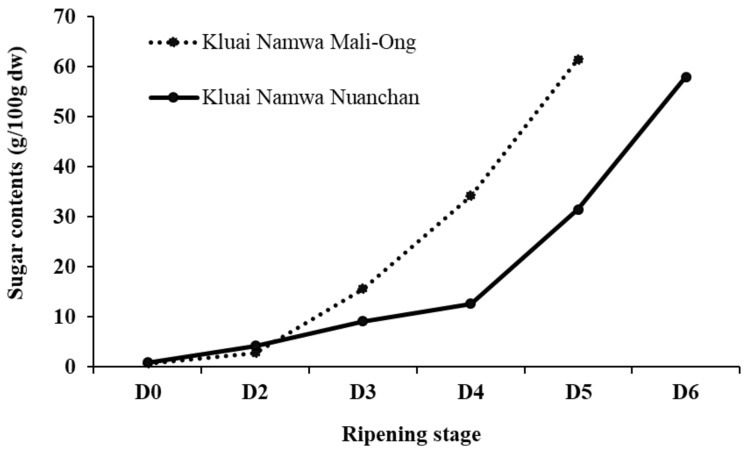
Total sugar contents (dried mass) of Namwa Mali Ong and Nuanchan varieties during the ripening stage.

**Figure 6 molecules-27-05675-f006:**
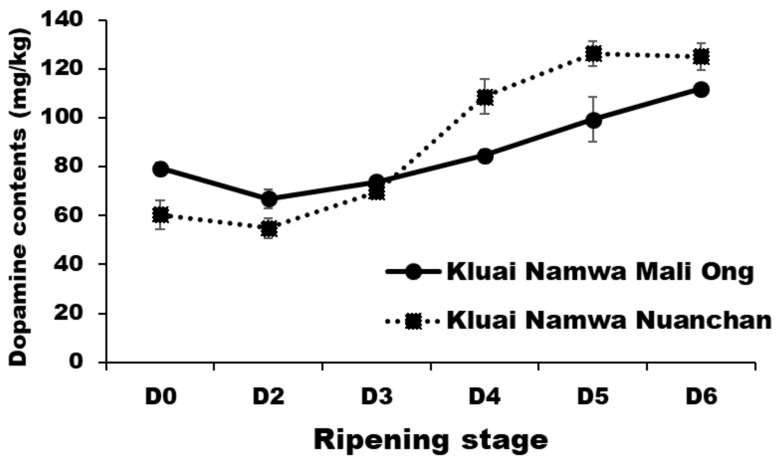
Dopamine contents (dry mass) of Namwa Mali Ong and Nuanchan varieties during the ripening stage.

**Figure 7 molecules-27-05675-f007:**
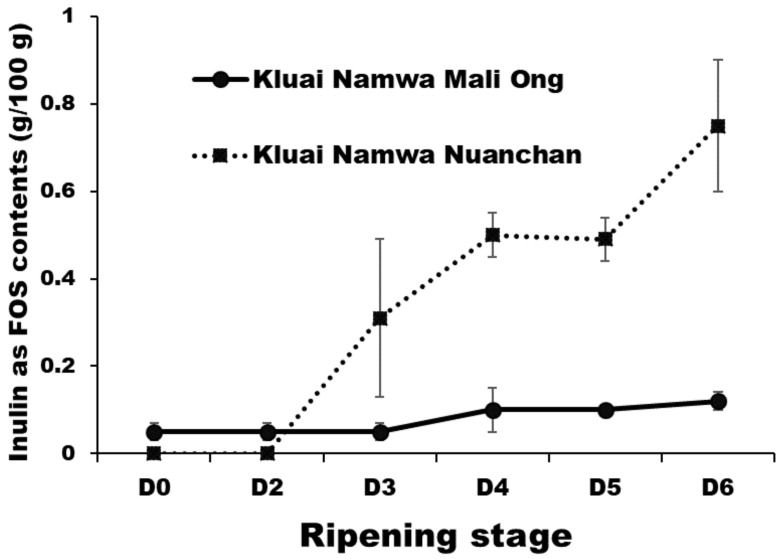
FOS contents (dry mass) of Namwa Mali Ong and Nuanchan varieties during the ripening stage.

**Figure 8 molecules-27-05675-f008:**
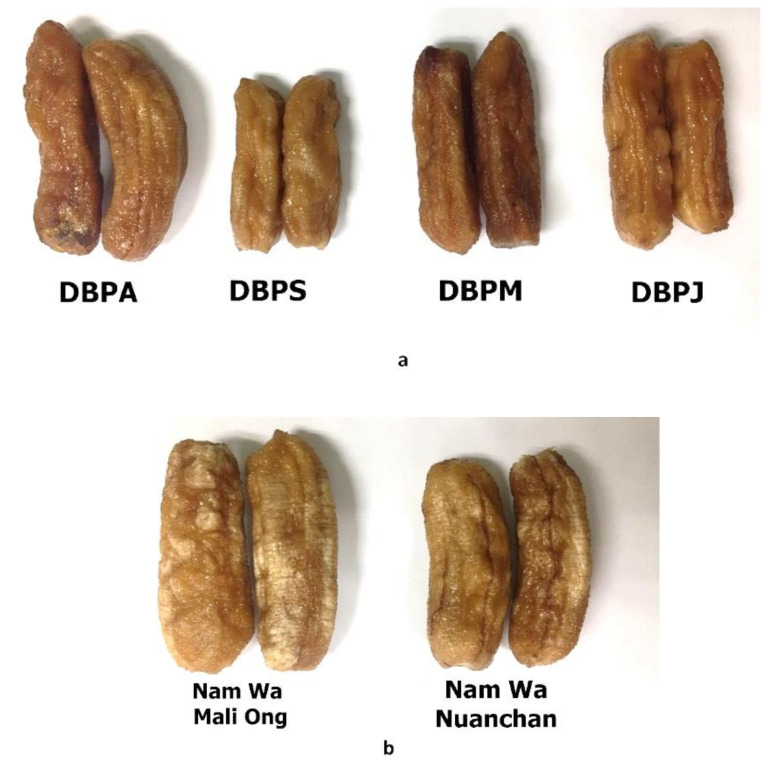
The four commercial sun-dried banana products (**a**) and sun-dried banana products made from Namwa Mali Ong and Nuanchan varieties (**b**).

**Figure 9 molecules-27-05675-f009:**
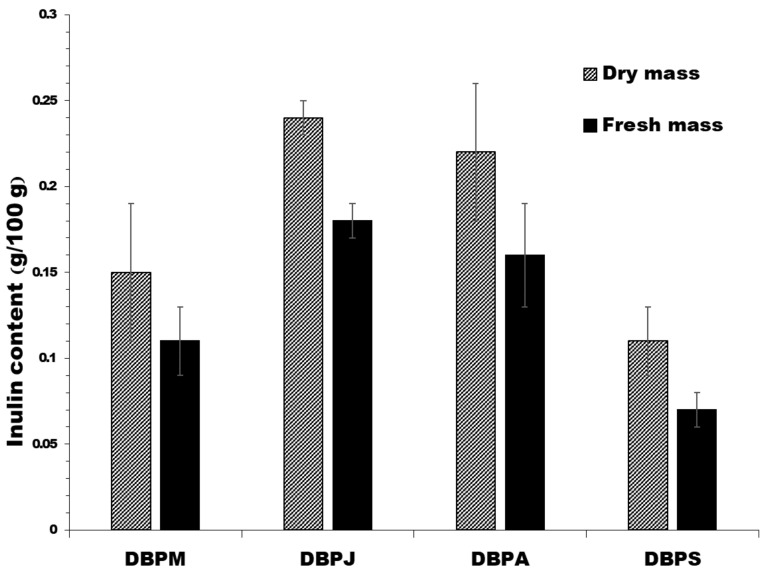
Inulin as FOS contents of the four commercial solar dried banana products.

**Figure 10 molecules-27-05675-f010:**
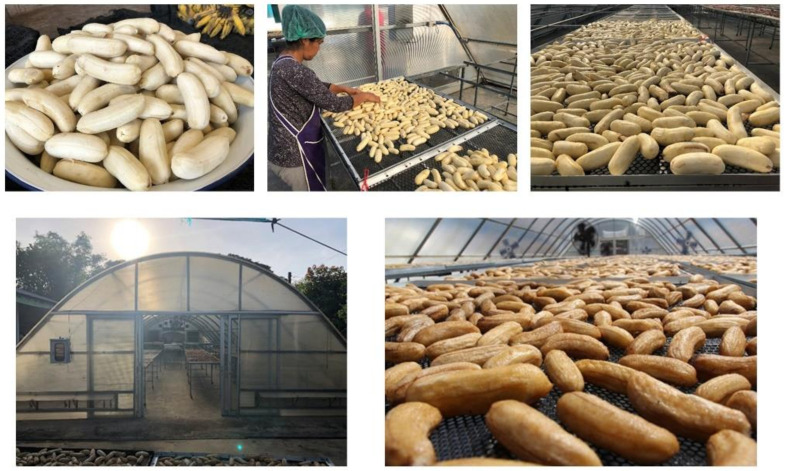
Processing of sun-dried banana products.

**Table 1 molecules-27-05675-t001:** Nutrient compositions of Kluai Namwa banana fruit and sun-dried banana products (basis per 100 g) [7].

Nutrient	Kluai Namwa (Raw Banana)	Sun-Dried Banana Product Made from Kluai Namwa	Unit
	Amounts	Amounts	
Energy	118	301	kcal
Water	68.8	19.9	g
Carbohydrates	27.18	67.0	g
Sugars	18.47	63.50	g
Dietary fiber	2.4	6.6	g
Protein	0.78	3.3	g
Fat	0.15	0.7	g
Vitamins			
Vitamin A (retinol activity equivalent)	3		mcg
B-carotene	39		mcg
Thiamine (B1)	0.04		mg
Riboflavin (B2)	0.03		mg
Niacin	0.99		mg
Vitamin C	13	5	mg
Minerals			
Calcium	7	3	mg
Magnesium	25		mg
Phosphorus	26		mg
Sodium	4	50	mg
Potassium	241		mg
Iron	0.52	0.8	mg
Copper	0.08		mg
Zinc	0.13		mg
Selenium	0.3		mcg

**Table 2 molecules-27-05675-t002:** Flavonoid contents (dry mass) of Namwa Mali Ong and Nuanchan cultivars at different ripening stages.

Ripening Stage	Flavonoids (mg/kg)							Total (mg/kg)
	Flavonols			Flavanone	Flavanols			
	Quercetin	Kampherol	Rutin	Naringenin	Catechin	Gallocatechin	Gallocatechin Gallate	
Kluai Namwa Mali Ong								
D0	61.82 ± 3.16 ^dA^	23.81 ± 3.98 ^cA^	157.40 ± 3.47 ^bA^	77.09 ± 1.35 ^eA^	77.22 ± 4.50 ^bA^	43.38 ± 2.38 ^aA^	18.74 ± 2.08 ^A^	459.44 ± 3.93 ^dA^
D2	96.73 ± 5.24 ^cA^	26.99 ± 2.23 ^bcA^	204.89 ± 6.70 ^aA^	116.42 ± 1.51 ^cA^	101.74 ± 8.87 ^aA^	25.99 ± 2.24 ^cA^	0	572.75 ± 22.31 ^abA^
D3	120.06 ± 0.76 ^bA^	29.54 ± 1.76 ^bB^	165.00 ± 0.71 ^bA^	103.83 ± 7.96 ^dA^	74.81 ± 4.16 ^bB^	32.95 ± 0.35 ^bA^	0	526.18 ± 8.55 ^cA^
D4	116.89 ± 4.24 ^bA^	28.60 ± 0.79 ^bcB^	165.79 ± 5.69 ^bA^	120.12 ± 3.07 ^cA^	97.67 ± 3.57 ^aA^	31.60 ± 3.42 ^bcA^	0	560.66 ± 2.26 ^bA^
D5	129.53 ± 3.97 ^aA^	35.74 ± 1.58 ^aA^	167.91 ± 4.89 ^bA^	134.87 ± 2.06 ^bA^	91.12 ± 4.04 ^aA^	35.86 ± 1.44 ^bA^	0	595.01 ± 5.77 ^aA^
D6	138.76 ± 3.84 ^aA^	35.75 ± 1.52 ^aA^	136.00 ± 3.91 ^cB^	147.23 ± 3.61 ^aA^	96.77 ± 6.27 ^aA^	30.18 ± 2.69 ^bcA^	0	584.67 ± 0.96 ^abA^
Kluai Namwa Nuanchan								
D0	71.97 ± 5.01 ^dA^	26.92 ± 1.94 ^eA^	202.49 ± 27.69 ^bA^	67.88 ± 5.31 ^cA^	99.84 ± 3.44 ^bA^	33.77 ± 2.38 ^aA^	21.40 ± 0.51 ^A^	524.27 ± 30.89 ^bA^
D2	98.01 ± 0.72 ^cA^	28.19 ± 0.48 ^deA^	154.35 ± 22.54 ^bcA^	114.36 ± 10.04 ^aA^	78.62 ± 3.13 ^dA^	31.65 ± 1.03 ^aA^	0	505.1715.81 ^bA^
D3	108.25 ± 9.01 ^cA^	33.96 ± 2.03 ^cdA^	129.15 ± 7.34 ^cA^	98.28 ± 6.75 ^bA^	87.80 ± 4.19 ^cA^	27.28 ± 0.98 ^bA^	0	484.71 ± 8.42 ^bA^
D4	121.87 ± 0.93 ^bA^	36.74 ± 0.85 ^cA^	260.38 ± 40.39 ^aA^	113.07 ± 3.10 ^aB^	97.21 ± 1.27 ^bA^	23.33 ± 0.70 ^bA^	0	652.59 ± 34.95 ^aA^
D5	135.00 ± 4.36 ^aA^	44.18 ± 5.06 ^bA^	187.39 ± 11.31 ^bcA^	115.77 ± 1.11 ^aA^	102.61 ± 3.45 ^abA^	16.94 ± 2.26 ^cA^	0	601.88 ± 14.31 ^aA^
D6	138.14 ± 3.27 ^aA^	50.37 ± 1.86 ^aA^	165.60 ± 5.86 ^bcA^	123.45 ± 2.84 ^aBA^	109.22 ± 2.34 ^aA^	15.65 ± 1.99 ^cA^	0	602.41 ± 7.64 ^aA^

Each value represents the mean ± S.D. For each variety, mean values within the column with different lowercase letter indicate significant differences (*p* < 0.05). For the same ripening stage between two banana varieties, values within the column with different uppercase letters indicate significant differences (*p* < 0.05).

**Table 3 molecules-27-05675-t003:** Phenolic acid contents (dry mass) of banana cultivars of Namwa Mali Ong and Nuanchan at different ripening stages.

Ripening Stage	Phenolic Acids (mg/g)					
	Tannic Acid	Gallic Acid	Ellagic Acid	Ferulic Acid	Chlorogenic Acid	Total (mg/g)
Namwa Mali Ong						
D0	112.02 ± 6.68 ^aA^	49.12 ± 1.51 ^cA^	96.60 ± 3.20 ^bA^	33.01 ± 3.03 ^aA^	47.09 ± 0.28 ^aA^	337.84 ± 11.67 ^aA^
D2	98.32 ± 3.04 ^bA^	32.81 ± 1.82 ^eA^	74.96 ± 5.19 ^cA^	20.86 ± 2.38 ^bA^	47.66 ± 2.86 ^aA^	274.61 ± 4.45 ^bA^
D3	84.07 ± 6.65 ^cA^	42.22 ± 1.78 ^dB^	81.37 ± 0.91 ^cA^	22.79 ± 1.78 ^bA^	45.33 ± 2.91 ^aB^	275.78 ± 8.65 ^bA^
D4	89.81 ± 0.64 ^bcA^	61.51 ± 1.22 ^bA^	114.31 ± 5.55 ^aA^	23.22 ± 2.80 ^bA^	44.76 ± 4.54 ^aB^	333.61 ± 0.06 ^aA^
D5	98.45 ± 6.70 ^bA^	69.77 ± 4.99 ^aA^	103.93 ± 3.60 ^bA^	21.31 ± 1.84 ^bA^	46.10 ± 0.98 ^aB^	339.56 ± 6.17 ^aA^
D6	90.33 ± 1.80 ^bcA^	51.20 ± 2.67 ^cB^	97.08 ± 1.53 ^bA^	0	45.22 ± 2.64 ^aA^	283.82 ± 0.30 ^bB^
Namwa Nuanchan						
D0	102.53 ± 5.61 ^bA^	45.44 ± 1.80 ^cB^	103.95 ± 1.55 ^abA^	19.75 ± 0.81 ^bcA^	50.12 ± 0.43 ^bA^	321.78 ± 4.12 ^bA^
D2	68.34 ± 2.69 ^dA^	46.42 ± 4.65 ^cA^	106.88 ± 4.79 ^abA^	17.70 ± 0.42 ^cA^	53.79 ± 1.99 ^bA^	293.13 ± 0.13 ^cA^
D3	67.02 ± 3.49 ^dA^	59.50 ± 2.53 ^bA^	112.07 ± 9.40 ^aA^	22.67 ± 1.44 ^abA^	52.45 ± 3.66 ^bA^	313.70 ± 3.36 ^bA^
D4	131.07 ± 10.51 ^aA^	56.39 ± 0.70 ^bA^	97.23 ± 3.44 ^bA^	22.14 ± 1.69 ^abA^	65.85 ± 3.94 ^aA^	372.66 ± 7.63 ^aA^
D5	96.14 ± 3.18 ^cA^	68.87 ± 3.45 ^aA^	67.27 ± 3.70 ^cB^	24.35 ± 1.01 ^aA^	66.56 ± 2.64 ^aA^	323.18 ± 6.68 ^bB^
D6	85.63 ± 0.86 ^cA^	69.00 ± 1.99 ^aA^	72.49 ± 3.95 ^cB^	20.54 ± 1.26 ^bc^	66.95 ± 1.21 ^aA^	314.60 ± 2.87 ^bA^

Each value represents the mean ± S.D. For each variety, mean values within the column with different lowercase letter indicate significant differences (*p* < 0.05). For the same ripening stage between two banana varieties, values within the column with different uppercase letters indicate significant differences (*p* < 0.05).

**Table 4 molecules-27-05675-t004:** Antioxidant value (dry mass) of Namwa Mali Ong and Nuanchan varieties during the ripening stages.

Ripening Stage	TPC (mg GAE/100 g)	Tannin (mg TAE/100 g)	TFC (mg QE/100 g)	DPPH (mg TE/100 g)	FRAP (mg FeSO_4_/100 g)
Kluai Namwa Mali Ong					
D0	160.01 ± 8.76 ^bB^	206.18 ± 10.86 ^bB^	176.06 ± 9.64 ^aA^	119.44 ± 12.45 ^dB^	970.28 ± 107.20 ^cA^
D2	195.61 ± 14.11 ^bB^	254.51 ± 25.57 ^bB^	97.30 ± 10.93 ^bB^	251.18 ± 22.79 ^cB^	2660.19 ± 140.09 ^bA^
D3	228.80 ± 15.57 ^bA^	287.66 ± 11.24 ^bA^	89.21 ± 6.44 ^bA^	437.30 ± 38.67 ^bA^	961.34 ± 104.33 ^cA^
D4	589.97 ± 25.71 ^aA^	801.38 ± 37.24 ^aA^	78.42 ± 11.80 ^bA^	732.40 ± 16.21 ^aA^	2392.84 ± 81.86 ^bA^
D5	601.38 ± 228.57 ^aA^	817.95 ± 266.34 ^aA^	157.03 ± 39.57 ^aA^	847.37 ± 232.27 ^aA^	5280.64 ± 1904.99 ^aA^
Kluai Namwa Nuanchan					
D0	207.30 ± 36.32 ^bcA^	274.97 ± 30.61 ^cdA^	184.29 ± 18.68 ^bA^	238.84 ± 49.56 ^dA^	917.74 ± 67.66 ^cA^
D2	728.57 ± 213.55 ^aA^	899.42 ± 261.50 ^aA^	211.62 ± 21.92 ^aA^	834.48 ± 161.08 ^bA^	2190.79 ± 191.58 ^aB^
D3	117.71 ± 29.72 ^cB^	145.69 ± 33.92 ^dB^	19.63 ± 8.87 ^dB^	50.89 ± 1.63 ^eB^	496.44 ± 179.09 ^dB^
D4	270.15 ± 25.60 ^bB^	378.46 ± 20.83 ^bcB^	82.33 ± 12.05 ^cA^	473.05 ± 29.82 ^cB^	1110.08 ± 134.81 ^bB^
D5	329.32 ± 18.03 ^bB^	459.16 ± 18.80 ^bB^	77.95 ± 3.95 ^cB^	1028.18 ± 28.32 ^aA^	1153.30 ± 51.45 ^bB^

TPC = total phenolic contents; TFC = total flavonoid contents; Each value represents the mean ± S.D. For each variety, mean values within the column with different lowercase letter indicate significant differences (*p* < 0.05). For the same ripening stage between two banana varieties, values within the column with different uppercase letters indicate significant differences (*p* < 0.05).

**Table 5 molecules-27-05675-t005:** Total sugar contents (g/100 g) dry mass of dried banana products made from Namwa Mali Ong and Nuanchan varieties and commercial products.

Samples	Fructose	Glucose	Sucrose	Total
Kluai Namwa Mali Ong				
Before drying	32.62 ± 0.37 ^a^	30.31 ± 0.29 ^a^	0	62.93 ± 0.10 ^a^
After drying	24.20 ± 0.06 ^b^	24.23 ± 0.16 ^b^	0.04 ± 0.03	48.47 ± 0.20 ^b^
Kluai Namwa Nuanchan				
Before drying	35.23 ± 0.13 ^a^	37.67 ± 0.18 ^a^	0	72.96 ± 0.09 ^a^
After drying	22.82 ± 0.12 ^b^	24.39 ± 0.94 ^b^	0	47.21 ± 1.05 ^b^
Commercial products				
DBPM	19.49 ± 0.11 ^c^	20.50 ± 0.03 ^d^	0	40.00 ± 0.14 ^d^
DBPJ	20.93 ± 0.16 ^b^	22.39 ± 0.06 ^b^	0	43.32 ± 0.11 ^b^
DBPA	23.05 ± 0.09 ^a^	24.93 ± 0.10 ^a^	0	47.98 ± 0.08 ^a^
DBPS	19.60 ± 0.05 ^c^	20.99 ± 0.08 ^c^	0.05 ± 0.01	40.65 ± 0.06 ^c^

Each value represents mean ± S.D. Mean values within the column for each variety with different superscript letter indicate significant differences (*p* < 0.05). For the four commercial products, values within the column with different superscript letter indicate significant differences (*p* < 0.05).

**Table 6 molecules-27-05675-t006:** Flavonoid contents (dry mass) of dried banana products made from Namwa Mali Ong and Nuanchan varieties and the four commercial products.

Samples	Nutraceuticals						
	Quercetin	Kampherol	Rutin	Naringenin	Catechin	Gallo- Catechin	Total mg/kg
Kluai Namwa Mali Ong							
Before drying	89.81 ± 1.01 ^a^	37.35 ± 2.83 ^a^	176.66 ± 6.92 ^a^	132.29 ± 3.99 ^a^	45.14 ± 3.42 ^a^	33.96 ± 3.80 ^a^	518.21 ± 8.13 ^a^
After drying	40.74 ± 0.87 ^b^ (55%)	16.55 ± 2.15 ^b^ (56%)	40.14 ± 2.37 ^b^ (77%)	31.18 ± 1.^72 b^ (76%)	35.59 ± 2.19 ^a^ (21%)	14.02 ± 1.64 ^b^ (59%)	178.21 ± 6.21 ^b^ (66%)
Kluai Namwa Nuanchan							
Before drying	110.54 ± 2.96 ^a^	28.67 ± 4.62 ^a^	166.39 ± 10.50 ^a^	82.41 ± 8.10 ^a^	84.84 ± 6.36 ^a^	23.71 ± 0.88 ^a^	496.54 ± 20.71 ^a^
After drying	39.53 ± 4.41 ^b^ (64%)	14.20 ± 1.24 ^a^ (50%)	54.23 ± 8 (67%)	43.34 ± 2.69 ^a^ (47%)	20.97 ± 1.53 ^a^ (75%)	9.85 ± 0.45 ^b^ (58%)	182.10 ± 1.35 ^b^ (63%)
The four commercial and two products from Mali Ong and Nuanchan							
DBPM	39.03 ± 2.10 ^b^	16.49 ± 1.65 ^ab^	38.34 ± 2.28 ^b^	45.22 ± 2.74 ^a^	22.18 ± 0.28 ^b^	10.42 ± 0.99 ^bc^	172.30 ± 5.30 ^c^
DBPJ	25.48 ± 3.37 ^c^	22.13 ± 4.21 ^a^	22.08 ± 2.29 ^c^	36.93 ± 3.14 ^cd^	32.66 ± 6.90 ^a^	9.16 ± 1.05 ^c^	148.44 ± 5.85 ^d^
DBPA	55.41 ± 3.06 ^a^	10.21 ± 1.62 ^c^	44.75 ± 1.16 ^ab^	40.66 ± 1.74 ^abc^	22.73 ± 2.03 ^b^	13.37 ± 1.38 ^abc^	187.14 ± 5.42 ^ab^
DBPS	55.72 ± 2.37 ^a^	12.92 ± 1.53 ^bc^	41.54 ± 2.65 ^b^	37.10 ± 2.44 ^bcd^	32.23 ± 1.63 ^a^	17.66 ± 3.52 ^a^	197.16 ± 1.45 ^a^
DBML	40.74 ± 0.87 ^b^	16.55 ± 2.15 ^ab^	40.14 ± 2.37 ^b^	31.18 ± 1.72 ^d^	35.59 ± 2.19 ^a^	14.02 ± 1.64 ^ab^	178.21 ± 6.20 ^bc^
DBNJ	39.96 ± 4.75 ^b^	14.20 ± 1.25 ^bc^	54.23 ± 8.70 ^a^	43.33 ± 2.69 ^ab^	20.97 ± 1.53 ^b^	9.85 ± 0.45 ^bc^	182.53 ± 1.97 ^bc^

DBML: products made from Mali Ong varieties, DBNJ: products made from Nuanchan varieties; The number (%) in parenthesis is the loss percentage after the drying process; Each value represents mean ± S.D. Values within the column for each variety with different superscript letter indicate significant differences (*p* < 0.05). For the four commercial products and two products from two varieties, values within the column with different superscript letter indicate significant differences (*p* < 0.05).

**Table 7 molecules-27-05675-t007:** Phenolic contents (dry mass) of dried banana products made from Namwa Mali Ong and Nuanchan varieties and the four commercial products.

Sample	Phenolic Acids					Total mg/kg
	Tannic Acid	Gallic Acid	Ellagic Acid	Ferulic Acid	Chlorogenic Acid	
Kluai Namwa Mali Ong						
Before drying	90.99 ± 3.63 ^a^	84.61 ± 5.29 ^a^	107.03 ± 2.83 ^a^	0	44.38 ± 4.41 ^a^	327.00 ± 0.38 ^a^
After drying	21.81 ± 2.31 ^b^ (76%)	22.17 ± 2.86 ^b^ (74%)	29.88 ± 1.79 ^b^ (72%)	0	22.19 ± 1.79 ^a^ (50%)	96.06 ± 0.55 ^b^ (70%)
Kluai Namwa Nuanchan						
Before drying	82.47 ± 8.73 ^a^	66.33 ± 3.52 ^a^	69.05 ± 2.91 ^a^	20.43 ± 1.69	47.84 ± 8.93 ^a^	286.12 ± 12.92 ^a^
After drying	32.35 ± 4.54 ^a^ (61%)	14.92 ± 3.39 ^a^ (76%)	36.38 ± 1.34 ^b^ (47%)	0 (100%)	18.54 ± 2.95 ^a^ (61%)	102.09 ± 0.46 ^b^ (64%)
The four commercial and two products from Mali Ong and Nuanchan						
DBPM	22.81 ± 2.08 ^cd^	18.35 ± 3.27 ^ab^	40.06 ± 2.64 ^b^	7.69 ± 0.95 ^b^	13.18 ± 2.54 ^bc^	102.09 ± 4.49 ^bc^
DBPJ	29.83 ± 0.83 ^bc^	14.07 ± 2.60 ^bc^	26.94 ± 3.97 ^c^	7.58 ± 0.02 ^b^	10.09 ± 0.79 ^c^	88.49 ± 0.23 ^d^
DBPA	32.03 ± 3.24 ^b^	18.42 ± 1.36 ^ab^	29.39 ± 1.44 ^c^	7.71 ± 0.89 ^b^	16.41 ± 2.82 ^b^	103.96 ± 2.36 ^b^
DBPS	46.49 ± 3.37 ^a^	8.79 ± 0.33 ^c^	46.29 ± 1.99 ^a^	10.09 ± 0.74 ^a^	14.42 ± 1.47 ^bc^	126.06 ± 3.48 ^a^
DBML	21.82 ± 2.31 ^d^	22.18 ± 2.86 ^a^	29.88 ± 1.79 ^c^	0	22.20 ± 1.79 ^a^	96.06 ± 0.54 ^c^
DBNJ	32.35 ± 4.54 ^b^	14.92 ± 3.39 ^bc^	36.39 ± 1.34 ^b^	0	18.54 ± 2.96 ^ab^	102.19 ± 0.47 ^bc^

DBML: products made from Mali Ong varieties, DBNJ: products made from Nuanchan varieties; The number (%) in parenthesis is the loss percentage after the drying process; Each value represents mean ± S.D. Values within the column for each variety with different superscript letter indicate significant differences (*p* < 0.05). For the four commercial products and two products from two varieties, values within the column with different superscript letter indicate significant differences (*p* < 0.05).

**Table 8 molecules-27-05675-t008:** Dopamine contents of commercial dried banana products and dried banana products from Namwa Mali Ong and Nuanchan varieties.

Sample	Dopamine (mg/kg)	Dopamine (mg/kg) (dw)
Nam Wa Mali Ong		
Before drying	30.39 ± 1.61 ^a^	101.44 ± 5.38 ^a^
After drying	18.00 ± 1.72 ^a^	24.52 ± 2.34 ^b^
Nam Wa Nuanchan		
Before drying	20.21 ± 2.86 ^a^	69.91 ± 9.91 ^a^
After drying	28.14 ± 1.29 ^a^	38.52 ± 1.76 ^a^
The four commercial and two products from Mali Ong and Nuanchan		
DBPM	24.37 ± 2.55 ^ab^	33.74 ± 3.54 ^ab^
DBPJ	26.52 ± 2.26 ^a^	35.87 ± 3.05 ^a^
DBPA	20.73 ± 2.12 ^bc^	28.59 ± 2.93 ^bc^
DBPS	15.20 ± 1.39 ^d^	22.26 ± 2.03 ^c^
DBML	18.00 ± 1.72 ^cd^	24.52 ± 2.34 ^c^
DBNJ	28.14 ± 1.29 ^a^	38.52 ± 1.76 ^a^

DBML: products made from Mali Ong varieties, DBNJ: products made from Nuanchan varieties; Each value represents mean ± S.D. Values within the column for each variety with different superscript letter indicate significant differences (*p* < 0.05). For the four commercial products and two products from two varieties, values within the column with different superscript letter indicate significant differences (*p* < 0.05).

**Table 9 molecules-27-05675-t009:** Antioxidant activities based on dry weight of commercial dried banana products and dried banana products made from Namwa Mali Ong and Nuanchan varieties.

Sample	TPC (mg GAE/100 g)	Tannin (mg TAE/100 g)	DPPH (mg TE/100 g)	FRAP (mg FeSO_4_/100 g)	TFC (mg QE/100 g)
Kluai Namwa Mali Ong					
Before drying	1276.74 ± 181.51 ^a^	23.82 ± 2.68 ^a^	5675.32 ± 266.95 ^a^	7593.33 ± 1670.99 ^a^	495.45 ± 93.66 ^a^
After drying	246.91 ± 16.25 ^b^ (81%)	4.95 ± 0.39 ^b^ (79%)	662.74 ± 14.65 ^b^ (88%)	1215.46 ± 50.02 ^b^ (84%)	120.78 ± 11.42 ^b^ (76%)
Kluai Namwa Nuanchan					
Before drying	737.75 ± 226.40 ^a^	14.98 ± 3.97 ^a^	3292.75 ± 155.58 ^a^	4404.39 ± 811.75 ^a^	255.04 ± 67.09 ^a^
After drying	249.22 ± 16.07 ^b^ (66%)	5.88 ± 0.28 ^b^ (61%)	632.33 ± 35.68 ^b^ (81%)	1280.07 ± 81.51 ^b^ (71%)	104.54 14.82 ^b^ (59%)
The four commercial and two products from Mali Ong and Nuanchan					
DBPM	266.05 ± 40.56 ^a^	5.33 ± 0.76 ^ab^	617.37 ± 45.92 ^a^	1110.58 ± 72.98 ^c^	44.71 ± 9.08 ^c^
DBPJ	286.62 ± 36.66 ^a^	5.84 ± 0.46 ^a^	643.85 ± 20.61 ^a^	1151.72 ± 34.13 ^bc^	45.03 ± 5.36 ^c^
DBPA	248.61 ± 62.23 ^a^	4.75 ± 0.92 ^b^	639.68 ± 29.73 ^a^	1129.19 ± 63.14 ^c^	44.82 ± 15.51 ^c^
DBPS	193.47 ± 19.77 ^b^	4.04 ± 0.19 ^c^	635.41 ± 19.63 ^a^	1108.55 ± 39.01 ^c^	59.16 ± 17.68 ^c^
DBML	246.91 ± 16.25 ^a^	4.95 ± 0.38 ^b^	662.74 ± 14.65 ^a^	1215.46 ± 50.02 ^ab^	120.79 ± 11.42 ^a^
DBNJ	249.22 ± 16.07 ^a^	5.88 ± 0.28 ^a^	632.33 ± 35.68 ^a^	1280.07 ± 81.51 ^a^	104.54 14.82 ^b^

DBML: products made from Mali Ong varieties, DBNJ: products made from Nuanchan varieties; The number (%) in parenthesis is the loss percentage after the drying process. Each value represents mean ± S.D. Values within the column for each variety with different superscript letter indicate significant differences (*p* < 0.05). For the four commercial products and two products from two varieties, values within the column with different superscript letter indicate significant differences (*p* < 0.05).

## Data Availability

Data are available from the authors on request.

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
