# Peer review of "Nutraceutical Difference between Two Popular Thai Namwa Cultivars Used for Sun Dried Banana Products"

_molecules, 2022, doi:10.3390/molecules27175675_

Round 1

Reviewer 1 Report

In the study, two varieties of banana were compared through analysis of different attributes before and after a sun drying process and as references for evaluation four commercial products were used. The study is basically a straightforward comparison, so that its strength is mainly data acquisition, hence the importance of precise performance of analysis of samples. Consequently, it is necessary to give better and precise description of how the samples were obtained for analysis. How was the selection performed? Which quantity of samples was taken from what population? Were the analysed samples representative? This could be considered a weakness of the present manuscript and correction is required. Otherwise, the results are well presented and discussed, under the assumption that sampling was performed correctly. The conclusion may also be considered relevant under the same assumption.

Specific comments:  

120: “... 4. more yellow than green, 5. more yellow than green, …” What is the difference between these two maturity stages? Line 120 differ from line 142 (142: “... stage 4 more yellow than green, stage 5 yellow with a trace of green, ...”)

243 “... narigenin, ...” Consider revising.

269: “Dopamine contents increased from 79.26 to 11.77 mg/kg ...” How can this be an increase?

275: “...between two varieties at each day of storage.” Consider revising.

502: “... dried in a parabolic dome at 50-60 °C for 4-5 days (Figure 9).” Is it Fig. 10, since Fig. 9 is a histogram for inulin content.

More than 2/3 of the references are articles published before 2012. Is there any possibility to update?

Author Response

In the study, two varieties of banana were compared through analysis of different attributes before and after a sun drying process and as references for evaluation four commercial products were used. The study is basically a straightforward comparison, so that its strength is mainly data acquisition, hence the importance of precise performance of analysis of samples. Consequently, it is necessary to give better and precise description of how the samples were obtained for analysis. How was the selection performed? Which quantity of samples was taken from what population? Were the analysed samples representative? This could be considered a weakness of the present manuscript and correction is required. Otherwise, the results are well presented and discussed, under the assumption that sampling was performed correctly. The conclusion may also be considered relevant under the same assumption.

Response: Thanks. The authors added some information in the Experimental design of banana ripening stage as in Line 498-503.

“Two varieties of green Kluai Namwa banana (ABB) (Musa sapientum L.) of Mali Ong and Nuanchan varieties were the freshly harvested in the early morning as clusters from a farm of Maesom Enterprise (Small and medium-sized enterprise) (Bang Krathum, Phitsanulok, Thailand) and immediately transferred to our laboratory (Phitsanulok). The harvesting period were 110 and 120 days after flowering for Mali Ong and Nuanchan varieties. The harvesting process was performed by Associated Professor Dr. Duangporn Premjet (botanist) and a farmer. Banana plantlet from Maesom farm were obtained from Pant Propagation Center No 6 (Phitsanulok). Clusters of each banana variety were separated into individual bunches of banana and stored at room temperature (30±0.81C) with relative humidity 40 ± 5.06%.”

Specific comments:  

120: “... 4. more yellow than green, 5. more yellow than green, …” What is the difference between these two maturity stages? Line 120 differ from line 142 (142: “... stage 4 more yellow than green, stage 5 yellow with a trace of green, ...”)

Response: Thanks. The authors changed “5. more yellow than green ” to “yellow with a trace of green (line 124)” as the suggestion.

243 “... narigenin, ...” Consider revising.

Response: Thanks. The authors changed narigenin to naringenin as the suggestion in Line#249.

269: “Dopamine contents increased from 79.26 to 11.77 mg/kg ...” How can this be an increase?

Response: It miss 1. The authors revised the dopamine contents from 11.77 mg/kg to 111.77 mg/kg in Line#276.

275: “...between two varieties at each day of storage.” Consider revising.

Response: The authors removed “ at each day of storage” (Line#281).

502: “... dried in a parabolic dome at 50-60 °C for 4-5 days (Figure 9).” Is it Fig. 10, since Fig. 9 is a histogram for inulin content.

Response: Thanks. The authors changed “... dried in a parabolic dome at 50-60 °C for 4-5 days (Figure 9).”  to “... dried in a parabolic dome at 50-60 °C for 4-5 days (Figure 10).” as suggestion in Line#519.

More than 2/3 of the references are articles published before 2012. Is there any possibility to update?

Response: Some old articles were selected as references due to similar methods were conducted in their studies. However, we’ve already replaced some updated articles here for references as suggestion in Line# 64 and 97 (ref No.5,14 and 15).

Reviewer 2 Report

In this manuscript, the authors investigated the nutraceutical contents of two popular Namwa varieties, Mali Ong and Nuanchan at different ripening stages. There are good application and science value. However, there are some significant issues that should be carefully analysed.

1) "p" in p < 0.5 should be italic in Line 43.

2) The authors do not point out the specific research significance of comparing the two kinds of bananas in this manuscript.

3) The authors thought that the two Namwa varieties could be used as replacements in short supply. Bananas are loved by people. In addition to quality factors, taste also plays an important role. Are there any significant differences between the two banana flavors?

4) There is a lack of difference analysis in Figure 9.

5) References are missing here in ‘3.4.2. HPLC and detector condition’.

Author Response

1) "p" in p < 0.5 should be italic in Line 43.

Response: Thanks. The authors revised as the suggestion.

2) The authors do not point out the specific research significance of comparing the two kinds of bananas in this manuscript.

Response: Thanks. The authors added some point in Line 71-73 and 76-77

3) The authors thought that the two Namwa varieties could be used as replacements in short supply. Bananas are loved by people. In addition to quality factors, taste also plays an important role. Are there any significant differences between the two banana flavors?

Response: For my point of view. I think that there is no flavor and texture difference at all both fresh consumption as banana fruit and the dried banana products. The farmers and the products producers also thinks the same idea as my point. However, the sensory evaluation is required for next study for your comment.

4) There is a lack of difference analysis in Figure 9.

Response: Thanks. The authors added analysis as in Line 437-440

5) References are missing here in ‘3.4.2. HPLC and detector condition’.

Response: The ref was added in Line 595 and detector was added in Line 587 and 592.